# A Systematic Bibliometric Analysis of the Real Estate Bubble Phenomenon: A Comprehensive Review of the Literature from 2007 to 2022

José-Francisco Vergara-Perucich

Núcleo Centro Producción del Espacio, Universidad de las Américas-Chile, Providencia 7500975, Chile;
jvergara@udla.cl

**Abstract:** This article presents the results of a bibliometric review of the study of real estate bubbles in the scientific literature indexed in Web of Science and Scopus, from 2007 to 2022. The analysis was developed using a sample of 2276 documents, which were reviewed in R software and analyzed with the assistance of the Bibliometrix package of the same software. The results indicate that there has been considerable productivity on the topic of real estate bubbles since 2007, with an emphasis on housing price formation processes and the social effects when bubbles burst. The authors found that there were not many case studies located in Latin America or Africa, nor were there approaches with advanced predictive modeling techniques using machine learning or artificial intelligence. The article provides an understanding of the state of the art in real estate bubble research and situates new research in front of the influential literature previously published.

**Keywords:** bibliometrics; real estate; bubble; housing prices; financial markets

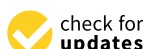



## 1. Introduction

In mid-2022, researchers at the Federal Reserve Bank of Dallas highlighted a concerning trend of a growing separation between house prices and fundamental variables that are typically taken into account to understand how such prices are formed. These changes were not observed since the 2007 crisis (Coulter et al. 2022). Lawrence Yun, the chief economist at the National Association of Realtors, emphasized the decoupling of income and housing affordability in an interview with CNN (Bahney 2022). Meanwhile, in China, a housing bubble has emerged, currently being addressed by the government (Brooker 2021), although the International Monetary Fund still recognizes it as a complex issue (Cheng 2023). Additionally, the Eurozone is grappling with a housing crisis characterized by soaring prices, managed through interest rates, but its efficacy remains uncertain (Read 2022). Concerns have also resurfaced regarding the dynamics of the residential market in Spain, with 2022 seeing flows similar to those observed prior to the 2007 crisis (ABC Madrid 2023). The recurring presence of the 2007 crisis in housing-related news worldwide underscores the formation of housing bubbles, where house prices significantly deviate over time from their fundamental values (Lind 2009).

While the 2008 crisis brought about renewed concerns in markets, global integration has not diminished, and associated risks remain latent (Li et al. 2021). Since then, specialized literature has made significant advances in areas related to real estate phenomena, for which the use of bibliometric techniques is not new. Jayantha and Oladinrin conducted a bibliometric study on hedonic prices using CiteSpace (Jayantha and Oladinrin 2019), while Li and Li conducted a bibliometric investigation on 60 years of housing prices (Li and Li 2022). Kong et al. conducted a bibliometric review related to urban sustainability research, incorporating insights from real estate (Kong et al. 2020), complemented by the study of Kaklauskas et al. (2021). Maggon conducted a specific thematic bibliometric study in the Journal of Corporate Real Estate (Maggon 2023). In an assessment of emerging trends from the pandemic, Naz

et al. conducted an analysis of the impact on property markets (Naz et al. 2022). Munawar et al. studied the impact of smart real estate on disaster management through the application of big data strategies (Munawar et al. 2020). In the field of investment and finance trends, there are also valuable contributions from bibliometric studies. Among the contributions is the bibliometric review on financial education by Goyal and Kumar (2021). The bibliometric study on green finance by Zhang et al. is also highly regarded (Zhang et al. 2019), as well as the research on management towards the United Nations Sustainable Development Goals, conducted by Pizzi et al. (2020). For the intersection of finance and real estate development, Zhao et al. conducted a study on green construction between 2000 and 2016, analyzing a total of 2980 articles (Zhao et al. 2019). Ganbat et al. conducted a study on BIM and risk, with a focus on international studies (Ganbat et al. 2018). The application of bibliometric techniques allows for the systematic analysis of extensive literature databases and provides a suitable overview for positioning new research in innovation and the generation of new knowledge.

This article aims to contribute to the understanding of this global concern through a bibliometric analysis of the literature on real estate price bubbles between the years 2007 and 2022. Conducting a bibliometric analysis allows us to present both quantitative and qualitative insights into the productivity associated with this field of economic research. The article begins by providing a general theoretical framework on real estate bubbles to situate readers within the significance of this research. Subsequently, the article explains the methods and techniques employed for the bibliometric analysis, leveraging freely available software such as R with the Bibliometrix package (Aria and Cuccurullo 2017). The data sources used for the analysis are obtained from Web of Science and Scopus, which provide comprehensive scientific production information. The results of the analysis are presented in the fourth section of the article, followed by a discussion of these findings and the conclusions drawn. This allows us to evaluate the limitations of this study and identify new research questions that emerge from the findings.

Himmerlberg et al. assert that a housing price bubble occurs when homeowners hold unrealistically high expectations of future earnings, leading them to perceive the current price as lower than the future price of homeownership (Himmelberg et al. 2005). However, it is important to note that not all abrupt increases in house prices constitute a housing bubble, as such price surges do not necessarily align with the definition of this phenomenon (Bourassa et al. 2019). According to Lind, bubbles are specific events characterized by a sustained increase in house prices followed by a sudden decline (Lind 2009). A precise definition of a price bubble involves a significant and expanding gap between the market value of an asset and the net present value of its dividends, indicating a detachment of the asset's value from the long-term growth of the economy in which it is traded (Domeij and Ellingsen 2018). Therefore, the persistent deviation of house price variations from the fluctuations in the costs of underlying fundamentals serves as an indicator of the existence of a real estate bubble (Stiglitz 2012).

In 2019, Bourassa et al. examined various approaches to measuring the presence of a house price bubble. They found that some authors identify bubbles by analyzing ratios comparing house prices to income or earnings. Additionally, researchers commonly employ regression analysis with different methodological approaches to explore the explanatory variables of price changes, and they may even draw upon techniques from physics to understand growth rates. In all cases, the aim is to anticipate the emergence of new bubbles due to their significant societal implications.

Brunnermeier identified the trigger for the bursting of the housing bubble in the United States as the unsustainable increase in mortgage lending coinciding with declining house prices, leading to a liquidity crisis and necessitating the design of a new financial architecture (Brunnermeier 2009). However, the full implementation of this new architecture remains pending. Since the occurrence of the financial crisis, academic groups and housing market experts have devoted their efforts to researching the 2007 crisis, gradually developing models to better anticipate the emergence of future housing bubbles. Although more than 2000 papers have been produced on this subject, organizing and synthesizing

this vast amount of information to identify knowledge gaps is a complex undertaking. This is where bibliometric exploration becomes valuable, aiding in the organization and classification of information.

The acceleration of scientific production makes it increasingly challenging to analyze the accumulated knowledge presented in a fragmented manner (Aria and Cuccurullo 2017). Bibliometric techniques provide a means to systematize this knowledge, employing repeatable techniques and transparent statistical strategies to understand the outcomes of scientific activity (Diodato 1994). By organizing scientific databases, bibliometrics enables the identification of underlying processes in knowledge production and facilitates the recognition of contributions to the field (Andres 2009; Sooryamoorthy 2021). Given the extensive volume of knowledge generated on real estate bubbles, this bibliometric study aims to contribute to the understanding of the current state of research in this field between 2007 and 2022.

## 2. Results

The results of the analysis focus exclusively on the period after the bursting of the real estate bubble in 2007 until 2022, when the annual production of scientific outputs did not fall below 50 and after 2015 productivity remained above 150 new articles per year, as shown in Figure 1. The most productive year was 2018.

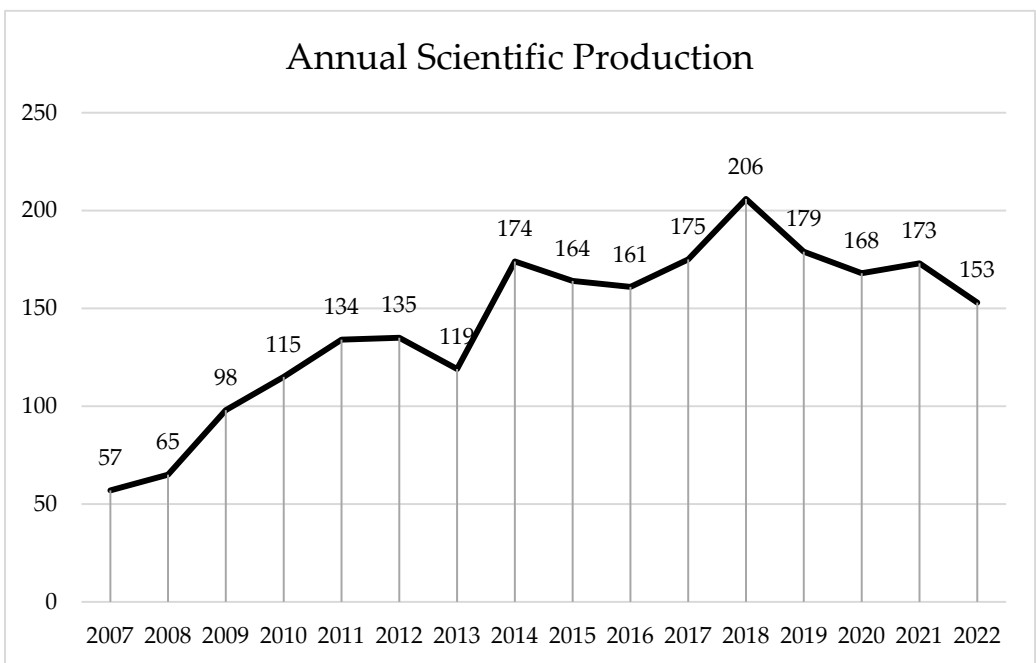

**Figure 1.** Annual productivity between 2007 and 2022. Source: Prepared by the authors with data from Web of Science and Scopus, assisted by the Bibliometrix program.

Figure 2 indicates three fields and their relationships with each other for this period. To the left of this graph, in the CR column are located the most commonly used cited references within the sample, generally referring to pre-crisis 2007 studies that sought to assess the presence of housing bubbles for different regions in the world. Within that sample, the most valuable text is Eddie C.M. Hui and Shen Yue's study of housing price bubbles in Hong Kong, Beijing, and Shanghai (Hui and Yue 2006). To the right of the figure are the keywords most commonly used by the authors to describe their articles on housing bubbles. This section shows the emphasis given to monetary policies and housing markets. In the center, the authors with the highest relative relevance within the sample are placed. The size of each block in each column indicates the importance of each factor for the sample. The next sections of this review will show how this graph is broken down into different analyses that offer new interpretations of the data.

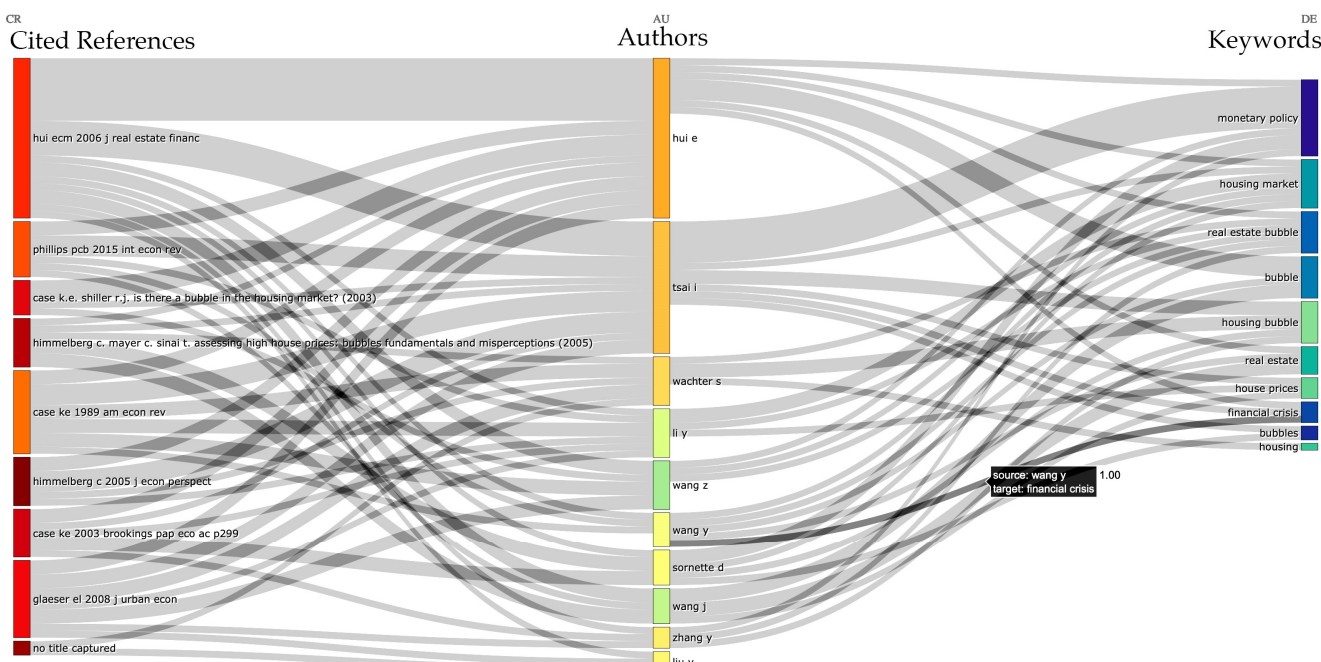

**Figure 2.** Sankey or Three-field plot between main references (**left**), authors (**center**), and keywords (**right**). Source: Own elaboration with data from Web of Science and Scopus, assisted by the Bibliometrix program.

Table 1 indicates the top 10 journals where articles can be found to inform opinion and review findings on real estate bubbles. This table ranks the journals according to their h-index, in order to assess not only the productivity but also the impact of each journal's publications. The Journal of Real Estate Finance and Economics stands out significantly, followed slightly behind by the Journal of Housing Economics.

**Table 1.** Top 10 journals ordered by H-Index measured by how many times the journal was cited by documents included in this sample. Source: Own elaboration with data from Web of Science and Scopus, assisted by Bibliometrix 4.1.4 software.

| # | Journal | H-Index | Times Cited | Articles on the Topic |
|---|---------|---------|-------------|-----------------------|
| 1 | Journal Of Real Estate Finance And Economics | 18 | 866 | 37 |
| 2 | Journal Of Housing Economics | 13 | 654 | 23 |
| 3 | Chemical Engineering Science | 12 | 477 | 15 |
| 4 | Habitat International | 11 | 363 | 14 |
| 5 | Real Estate Economics | 11 | 443 | 23 |
| 6 | Economic Modelling | 10 | 300 | 18 |
| 7 | Journal Of Real Estate Research | 9 | 282 | 15 |
| 8 | Physica A-Statistical Mechanics And Its Applications | 9 | 196 | 15 |
| 9 | Housing Studies | 8 | 278 | 14 |
| 10 | International Journal Of Housing Markets And Analysis | 8 | 213 | 31 |

Along the same lines of reviewing the impact of publications, the 10 most influential articles on real estate bubbles are identified, measured by the number of citations per year (Table 2). As these are the most cited, an abridged version of their abstracts has been included in the table to better understand what each article was about. Brunnermeier's paper was published just after the 2007 bubble and is by far the most influential. In this

document, the author explains the economic mechanisms that led to the great crisis, a few years after it occurred, in a didactic way and with a descriptive value relevant to any text that seeks to understand this phenomenon (Brunnermeier 2009). Schiller's book, an expert on price bubbles, is also highly cited, and attempts to provide a precautionary view of the development of new crises similar to those of 2007, indicating the elements that could establish similar scenarios (Shiller 2015). The text by Rugh and Massey presents evidence that indicates that residential segregation allows us to geographically understand the patterns of development of the crisis, given that segregated households are more dependent on mortgage loans to purchase homes and, in the face of the financial crisis, these households, being dependent on the good return on capital, are also the first to suffer foreclosures when they are unable to pay their mortgage repayments (Rugh and Massey 2010).

In this selection of influential articles ordered by citations by year, two contributions by Edward Glaeser appear. Firstly, together with Huang, Ma, and Shleifer, he presents evidence on how a price bubble operates in the Chinese housing market, that despite the existence of a bubble, the high demand for housing makes the price increase sustainably, so that central government stabilization is not necessary (Glaeser et al. 2017). The second article contributed by Glaeser is closer to the bursting of the bubble in 2007, where he and Gyourko and Sainz demonstrate the importance of the elasticity of supply in the housing market, which allows the economic effects of bubbles to be less long-lasting (Glaeser et al. 2008). The most influential papers are generally characterized by methodological applications of advanced statistical techniques that seek to review the relationships of housing shocks with other factors such as credit, housing markets, and fundamentals.

Figure 3 illustrates the progress of the keywords of the articles in the period studied. This analysis allows us to understand how the priorities of researchers in the study of real estate bubbles have changed. As expected, the first years after the bursting of the housing bubble focused on terms such as recession, risk, financial crisis, and monetary policy. By 2016, when productivity increases by more than 150 articles per year on these topics, housing prices appear in a sustained manner over time. In the closest period recorded, Granger causality appears as a keyword, which indicates that the methodological approach to the study of the phenomenon becomes more prevalent in the studies, something that also occurred in the 2011 and 2014 articles, when panel data and cointegration analysis were used predominantly, respectively.

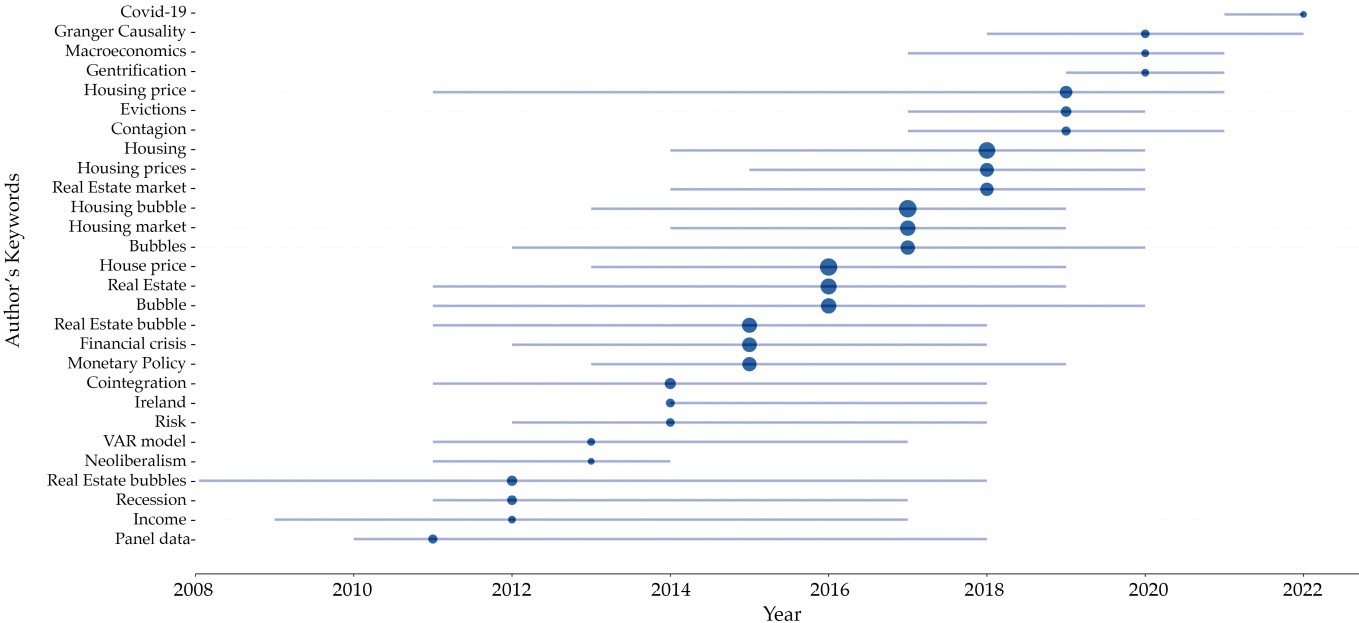

**Figure 3.** Authors' keyword trends over time. Source: Own elaboration with data from Web of Science and Scopus, assisted by Bibliometrix 4.1.4 software.

**Table 2.** Most influential papers cited at a global level (including documents excluded from this sample) ordered by times cited. Source: Own elaboration with data from Web of Science and Scopus, assisted by Bibliometrix 4.1.4 software.

| # | Short Reference | Total Citations per Year | Abstract | DOI |
|---|---|---|---|---|
| 1 | BRUNNERMEIER MK, 2009, J ECON PERSPECT | 91.27 | This paper attempts to explain the economic mechanisms that caused losses in the mortgage market to amplify into such large dislocations and turmoil in the financial markets, and describes common economic threads that explain the plethora of market declines, liquidity dry-ups, defaults, and bailouts that occurred after the crisis broke in summer 2007. (Brunnermeier 2009) | 10.1257/jep.23.1.77 |
| 2 | SHILLER RJ, 2015, IRRATIONAL EXUBERANCE | 32.78 | With high stock and bond prices and the rising cost of housing, the post-subprime boom may well turn out to be another illustration of Shiller's influential argument that psychologically driven volatility is an inherent characteristic of all asset markets. In other words, Irrational Exuberance is as relevant as ever. Previous editions covered the stock and housing markets—and famously predicted their crashes. (Shiller 2015) | Book ISBN = 9780691173122 |
| 3 | RUGH JS, 2010, AM SOCIOL REV | 31.36 | We argue that residential segregation created a unique niche of minority clients who were differentially marketed risky subprime loans that were in great demand for use in mortgage-backed securities that could be sold on secondary markets. We test this argument by regressing foreclosure actions in the top 100 U.S. metropolitan areas on measures of black, hispanic, and Asian. We find that black residential dissimilarity and spatial isolation are powerful predictors of foreclosures across U.S. metropolitan areas. We thus conclude that segregation was an important contributing cause of the foreclosure crisis. (Rugh and Massey 2010) | 10.1177/0003122410380868 |
| 4 | GLAESER E, 2017, J ECON PERSPECT | 25.43 | The demand for real estate in china is so strong that current prices might be sustainable, especially given the sparse alternative investments for Chinese households, so long as the level of new supply is radically curtailed. Whether that happens depends on the policies of the Chinese government, which must weigh the benefits of price stability against the costs of restricting urban growth. (Glaeser et al. 2017) | 10.1257/jep.31.1.93 |
| 5 | GLAESER EL, 2008, J URBAN ECON | 23.69 | In this paper, we present a simple model of housing bubbles that predicts that places with more elastic housing supply have fewer and shorter bubbles, with smaller price increases. The data show that the price run-ups of the 1980s were almost exclusively experienced in cities where housing supply is more inelastic. More elastic places had slightly larger increases in building during that period. (Glaeser et al. 2008) | 10.1016/j.jue.2008.07.007 |
| 6 | PHILLIPS PCB, 2011, QUANT ECON | 21.38 | Three relevant financial series are investigated, including a financial asset price (a house price index), a commodity price (the crude oil price), and one bond price (the spread between baa and aaa). Statistically significant bubble characteristics are found in all of these series. The empirical estimates of the origination and collapse dates suggest a migration mechanism among the financial variables. Our empirical estimates of the origination and collapse dates and tests of migration across markets match well with the general dateline of the crisis put forward in the recent study by Caballero et al. (2008). (Phillips and Yu 2011) | 10.3982/QE82 |
| 7 | MARTIN R, 2011, J ECON GEOGR | 19.62 | The recent financial crisis, with its origins in the collapse of the sub-prime mortgage boom and house price bubble in the USA, is a shown to have been a striking example of 'glocalisation', with distinctly locally varying origins and global consequences and feedbacks. The shift from a 'locally originate and locally-hold' model of mortgage provision to a securitised 'locally originate and globally distribute' model meant that when local subprime mortgage markets collapsed in the USA, the repercussions were felt globally. (Martin 2011) | 10.1093/jeg/lbq024 |

**Table 2.** *Cont.*

| # | Short Reference | Total Citations per Year | Abstract | DOI |
|---|---|---|---|---|
| 8 | WU J, 2012, REG SCI URBAN ECON | 18.58 | Our calculations suggest that even modest declines in expected appreciation would lead to large price declines of over 40% in markets such as Beijing, absent offsetting rent increases or other countervailing factors. Price-to-income ratios also are at their highest levels ever in Beijing and select other markets, but urban income growth has outpaced price appreciation in major markets off the coast. Much of the increase in prices is occurring in land values. (Wu et al. 2012) | 10.1016/j.regsciurbeco.2011.03.003 |
| 9 | JORDA O, 2015, J MONETARY ECON | 18 | What risks do asset price bubbles pose for the economy? This paper studies bubbles in housing and equity markets in 17 countries over the past 140 years. We demonstrate that what makes some bubbles more dangerous than others is credit. When fueled by credit booms, asset price bubbles increase financial crisis risks; upon collapse they tend to be followed by deeper recessions and slower recoveries. (Jordà et al. 2015) | 10.1016/j.jmoneco.2015.08.005 |
| 10 | AALBERS M, 2009, AREA | 16.13 | This article does not present new empirical research, but brings together work from different literatures that all in some way have a specific angle on the financial crisis. The aim of this article is to make the geographical dimensions of the financial crisis understandable to geographers that are not specialists in all—or even any—of these literatures, so that they can comprehend the spatialisation of this crisis. (Aalbers 2009) | 10.1111/j.1475-4762.2008.00877.x |

In relation to the productivity associated with the study of real estate bubbles, the dominant nations are by an overwhelming majority the United States of America and China (Table 3). In both cases, it is observed that they were nations exposed to bubbles, in the case of the United States of America with the bursting of the bubble in 2007 and in the case of China with the threat of a bubble that has been followed since 2005, for which in 2021 there was a potential risk of bursting that finally did not materialize. Spain and the United Kingdom have similar levels of productivity in this area. In this selection of the nations that contribute the most research results, there are no Latin American or African nations. The lack of research on real estate bubbles in Latin America may stem from the relatively lower occurrence of major housing market crises in the region compared to the United States and China. The bursting of the housing bubble in 2007 had a profound impact on the research agenda in the United States, while the threat of a bubble in China has attracted considerable attention. Latin American countries may not have experienced similar levels of housing market volatility, leading to less research emphasis in this area. Additionally, access to comprehensive databases and scholarly publications, such as the Web of Science and Scopus, which were used in the analysis, might be more restricted in Latin American research institutions.

**Table 3.** Country Scientific Production. Source: Own elaboration with data from Web of Science and Scopus, assisted by Bibliometrix 4.1.4 software.

| # | Country | Number of Documents |
|---|---|---|
| 1 | United States of America | 757 |
| 2 | China | 653 |
| 3 | Spain | 270 |
| 4 | United Kingdom | 224 |
| 5 | Germany | 155 |
| 6 | Japan | 95 |
| 7 | Canada | 88 |
| 8 | Australia | 87 |
| 9 | Italy | 83 |
| 10 | India | 78 |
| 11 | France | 69 |
| 12 | Turkey | 58 |
| 13 | South Korea | 57 |
| 14 | Netherlands | 54 |
| 15 | Czech Republic | 50 |

Figure 4 elaborates on the aggregate contents of productivity in real estate bubbles. This figure is a thematic map divided into four areas: Niche themes, which are developed by a specific group of authors; driving themes, which have progressively increased productivity over time; emerging or declining themes, which show significant variations in productivity abruptly and more recently; and core themes, which are present in most articles. Among the core themes, housing prices and financial crisis are mainly observed. On the borderline between core and driving themes are housing crises, foreclosures, evictions, and financialization, which are topics that have increased productivity in this area of study. Among the niche topics are the concepts of crashes, possibly associated with risk assessment. Contract for Difference also appears as an investment instrument with real estate backing, and in the area of topics that have presented recent variations, consumption appears.

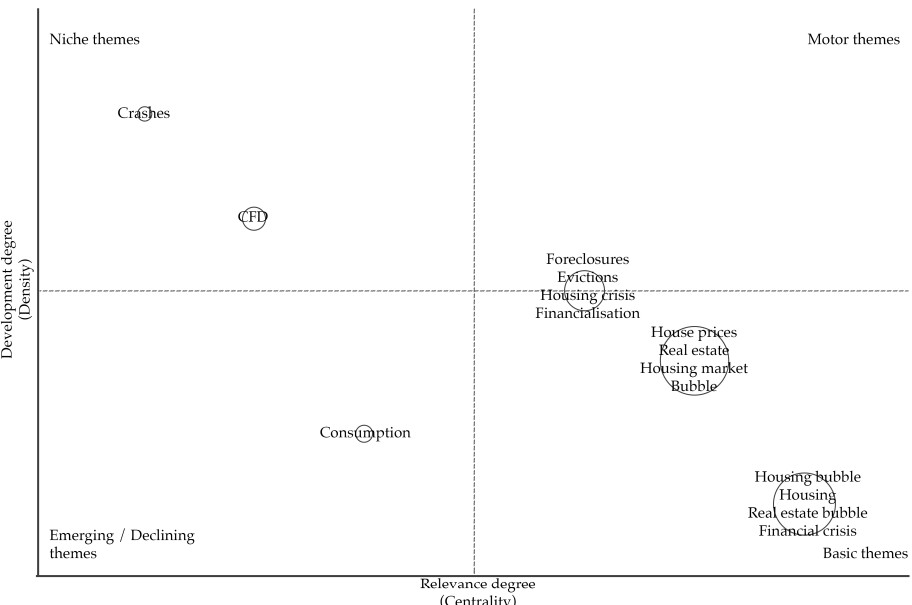

**Figure 4.** Thematic map of author's keywords. Source: own elaboration with data from Web of Science and Scopus, assisted by Bibliometrix 4.1.4 software.

Figure 5 corresponds to a word cloud with the most repeated trigrams in the abstracts of the reviewed publications. A trigram is the use of three concepts in an established order, excluding stopwords. In part, the trigrams allow us to complement the analysis of the thematic maps previously presented, with information that is in the abstracts, where usually the authors can develop the focus, scope and results of their documents. The focus of the articles reviewed indicates that there is a prevalence for studying housing prices, but it is also observed that there are methodological elements that speak of searching for causal results based on time series, such as the concepts of Granger causality tests or unit root tests. The Chinese real estate market has a special presence in this word cloud, as does real estate investment. Figure 6 shows a synthesis of the flows of collaborations between researchers from different countries, differentiated by thematic group and common sources according to the color of each cluster.

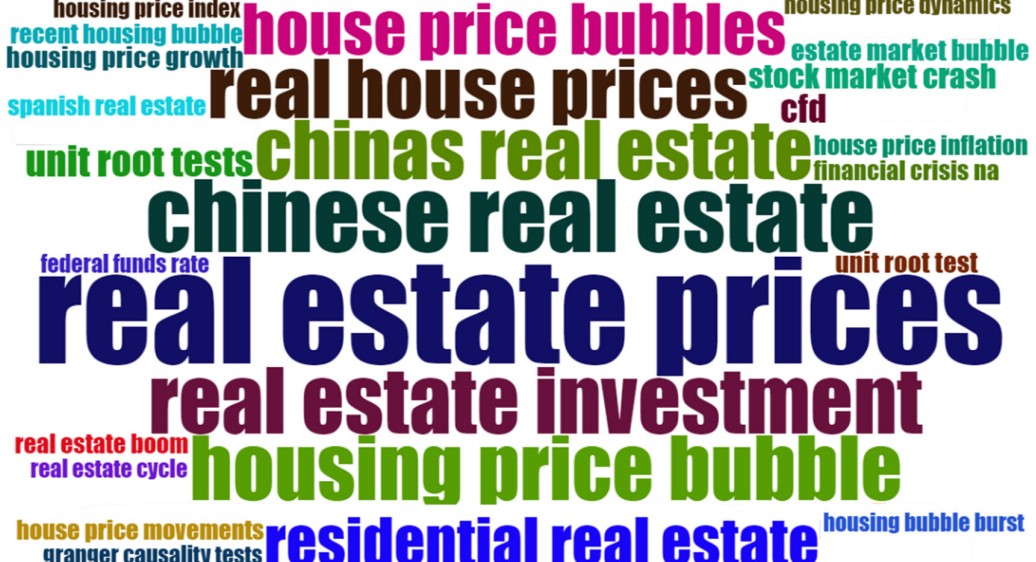

**Figure 5.** Wordcloud of trigrams from abstracts. Source: Own elaboration with data from Web of Science and Scopus, assisted by Bibliometrix 4.1.4 software.

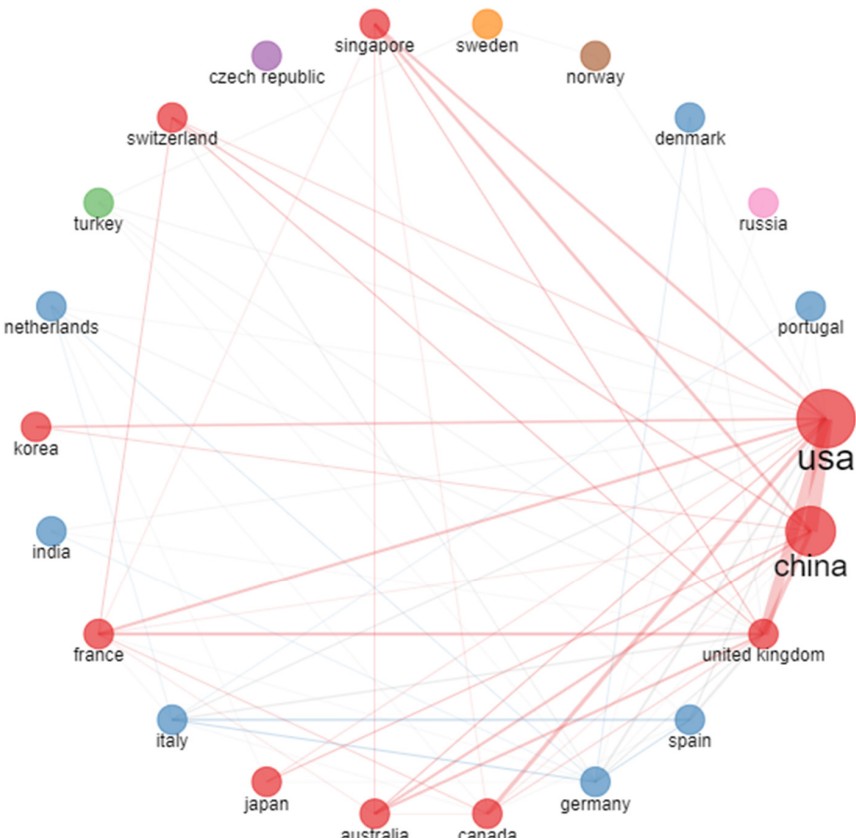

**Figure 6.** Collaboration network between countries. Source: Own elaboration with data from Web of Science and Scopus, assisted by Bibliometrix 4.1.4 software.

Finally, there is a general consensus that the origin of the bubble was a combination of deregulation of housing financial markets, innovation in financial instruments regarding their migration models to other instruments, and the disconnection between housing prices and their fundamentals, resulting in artificial values far from their true exchange values.

### 3. Discussion

Since the 2007 crisis, there has been a growing concern among the scientific community, as well as experts in finance and real estate markets, about the potential emergence of new price bubbles, particularly in the housing market. The number of papers addressing this topic has increased significantly, with a 755% rise in productivity from 2007 to 2022. However, despite the dynamic nature of knowledge today, the thematic variations observed during this period have not been as diverse as expected. Figure 5 illustrates this phenomenon, with only consumption appearing as an emerging topic. Surprisingly, aspects such as artificial intelligence, machine learning, and smart cities are notably absent from new approaches in this field of study. This lack of thematic innovation over the 15-year period may be attributed to a certain level of conformity with existing methodologies used to monitor the sustainability of the real estate market. Alternatively, these innovative approaches may still be relatively unknown to experts in the field.

The motivation for incorporating these advanced methodologies is justified because machine learning and AI algorithms can process and analyze large volumes of data more efficiently than traditional statistical methods. Real estate markets generate vast amounts of complex data, including property prices, transaction records, economic indicators, and social factors. Applying machine learning techniques can help identify patterns, detect anomalies, and uncover hidden relationships within these datasets, enabling a deeper understanding of the factors contributing to the formation and evolution of real estate bubbles. Also, machine learning algorithms have the potential to improve the accuracy of predictive models for real

estate bubbles. By training models on historical data, machine learning algorithms can learn from past patterns and behaviors to forecast future market trends and identify early warning signs of potential bubbles. This can assist policymakers, investors, and market participants in making informed decisions and implementing timely interventions to mitigate the risks associated with housing market crises. Also, there is the issue of nonlinear relationships as real estate bubbles may exhibit nonlinear dynamics and complex interactions among various factors. Machine learning techniques, such as neural networks and support vector machines, are capable of capturing nonlinear relationships and identifying non-obvious connections between variables.

In addition to the lack of thematic innovation, the research on real estate bubbles also lacks geographical diversity. The majority of the research has been conducted in developed countries, with little attention paid to emerging markets such as Latin America and Africa. This is a significant oversight, as we do not actually know if these regions are particularly vulnerable to the effects of real estate bubbles, as there has not been enough research conducted in these regions.

The analysis reveals the presence of three distinct bodies of production categorized by their approaches. First, there is a purely financial approach that aims to understand the rationality behind the formation of real estate bubbles, with the intention of influencing decision-makers to prevent market collapses. Second, there is a focus on methodological aspects, seeking to explore empirical techniques that enhance the accuracy in measuring the formation of real estate bubbles. Third, there is an approach that examines the social causes and consequences of real estate bubbles, with a particular concern for social justice and equity. All three approaches hold value and are necessary for a comprehensive understanding of the subject, but they often lack interaction with one another.

The findings of this study have implications for policy makers. As mentioned, the research on real estate bubbles has important implications for the design of financial regulation and housing policy. Policy makers need to be aware of the factors that contribute to the formation of bubbles and the potential consequences of these bubbles. They also need to be aware of the limitations of existing methods for monitoring and preventing bubbles.

Additionally, it has been observed that certain nations tend to collaborate with each other in producing research papers on housing bubbles. Figure 7 suggests a potential categorization of nations based on their emphasis on liberal approaches (represented by the red group), welfare states (represented by the blue group), or local and less associated approaches. To gain further insights, it may be worthwhile to conduct separate bibliometric analyses for these groups of nations, examining how the results vary and potentially confirming the hypothesis of distinct associations with liberal or social democratic outlooks.

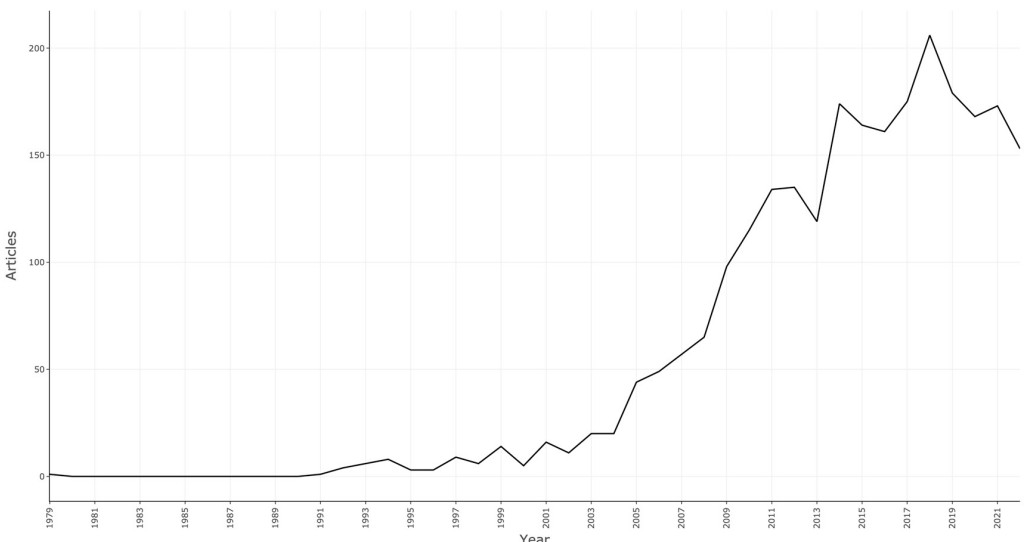

**Figure 7.** Annual productivity between 1979 and 2022.

Finally, there is a need for more thematic and methodological innovation in this field. Researchers could develop new methods for monitoring and preventing bubbles, and they need to explore the potential for using artificial intelligence and machine learning techniques to address these issues with more advanced and comprehensive tools.

In conclusion, this study employed bibliometric analysis to gain a systematic understanding of scientific productivity concerning real estate bubbles, utilizing data obtained from renowned sources of scientific information, such as the Web of Science and Scopus. The analysis encompassed a review of 2276 documents spanning from 2007 to 2022. The findings shed light on the most influential articles and the key concepts that elucidate the research trends within this field of study.

Several noteworthy considerations arise from the study. First, it is surprising to observe the absence of dominant studies from Latin America and Africa, regions facing housing crises due to soaring prices. These regions could greatly benefit from exploring real estate bubbles as a dedicated field of study. Moreover, the analysis indicates a lack of thematic or methodological innovation in the majority of studies, especially the most influential ones. It is imperative to recognize the potential for utilizing artificial intelligence and machine learning techniques to address these issues. While these technologies may still be in their early stages, it is highly appropriate to develop automated mechanisms capable of monitoring and preventing the emergence of bubbles. The required technology and capabilities already exist, awaiting broader implementation in this domain.

## 4. Materials and Methods

This study takes a mixed approach to evaluate the scope of the reviewed literature through a bibliometric analysis of sources using a quantitative way of organizing qualitative information and also statistics from indexed journals (Campbell 2011; Singleton 2010). The aim of this analysis is to systematize the available information on real estate bubbles, identifying gaps in knowledge and generating questions that allow for further exploration of areas of research that require more information (Moslehpour et al. 2022). The data for this analysis are obtained from the two main peer-reviewed scientific productivity databases currently available: Web of Science and Scopus (Cascajares et al. 2021; Xie et al. 2020). Both sources bring together the highest-impact scientific publications and the most prestigious international journals in the world. In both cases, advanced searches for terms were applied, as indicated below:

*Web of Science: TS(Housing OR House OR Real Estate) AND TS(Bubble)*
*SCOPUS: ALL(Housing OR House OR Real Estate) AND ALL(Bubble)*

The search results yielded 2496 results ranging from 1979 to 2023, with 1336 different sources (journals, books, and conferences), with an average annual growth rate in productivity of 12.41%, with documents that, on average, are cited 13.82 times and that, in turn, have cited 86,160 different references (Table 4). Applying a time filter to this sample to consider only publications since the year of the global financial crisis in 2007, the number of papers is reduced to 2276, with an annual growth rate of 6.8%, indicating that there has been a contraction in productivity associated with this problem, but as can be seen in Figure 7, this is mainly due to strong growth after the 2007 crisis, when it goes from 49 results in 2006 to 115 in 2010. After the crisis, there were never fewer than 100 publications per year on the subject.

The data obtained from Web of Science and Scopus were merged using the R package called bibliometrix, with its interface called *biblioshiny.* Although Scopus and Web of Science (WoS) are widely recognized and reputable databases, it is crucial to acknowledge that there are variations in the quality of data provided by each platform. It should be noted that some journals are indexed in both databases, which can lead to potential errors, duplications, or inaccuracies in the data. These issues can ultimately impact the integrity of the analysis results. Therefore, to address this concern, a data pre-processing step was implemented, which involved comprehensive deduplication using R and the DPLYR package. Subsequently, a visual review of the data cleansing was conducted. As part of this pre-processing stage, approximately 830 duplicated records were eliminated from the sample.

**Table 4.** Main information of data collected. Source: Own elaboration based on Web of Science and Scopus.

| Timespan | 1979–2022 | 2007–2022 |
|---|---|---|
| Sources | 1336 | 1224 |
| Documents | 2496 | 2276 |
| Annual Growth Rate % | 12.41 | 6.8 |
| Document Average Age | 8.59 | 7.4 |
| Average citations per doc | 13.82 | 12.19 |
| References cited | 86,160 | 81,704 |

There are some advantages of using the bibliometrix package (Moral-Muñoz et al. 2020; Linnenluecke et al. 2020; Munim et al. 2020). It provides a flexible framework for bibliometric analysis, allowing researchers to customize and adapt the way of conducting the study according to their specific research questions and needs. It offers a wide range of functions and parameters that can be tailored to the analysis objectives such as data collection and preprocessing in the R environment to tidy up the datasets. Also, the package offers network analysis capabilities, allowing researchers to analyze collaboration networks, co-citation networks, and co-word networks, not only as visualization as in VOSviewer, but also based on its indicators and functions to calculate network centrality measures, identify clusters or communities within the network, and visualize network structures in a similar way to VOSviewer. Furthermore, the use of temporal analysis enables researchers to examine temporal trends in bibliometric data. It offers functions to identify prolific authors, assess author productivity and collaboration patterns, and explore the research performance of institutions, and also use keywords and abstract n-grams to elaborate more complex analysis near to text-mining tools (which may also be incorporated from the R environment although it was not applied to this research particularly). So, as a package in R, bibliometrix seamlessly integrates with other R packages and tools, enabling researchers to incorporate bibliometric analysis into their existing R workflows. This integration facilitates data preprocessing, statistical analysis, and visualization, as R offers a comprehensive ecosystem of packages for these tasks. Finally, bibliometrix facilitates the automation of the entire bibliometric analysis process, from data collection to visualization, using R scripts. This enables reproducibility, as the analysis can be easily replicated and shared with others, enhancing transparency and facilitating collaboration.

## 5. Conclusions

The present study employed a comprehensive bibliometric analysis to examine the scientific productivity and research trends related to real estate bubbles. By utilizing data from reputable sources such as the Web of Science and Scopus, a thorough analysis of 2276 documents published between 2007 and 2022 was conducted. The findings provide valuable insights and lead to several noteworthy conclusions.

Firstly, it is evident that the concern surrounding the potential emergence of real estate bubbles, particularly in the housing market, has been a focal point for the scientific community and experts in finance and real estate markets since the 2007 crisis. The substantial increase in productivity, as indicated by the 755% rise in the number of papers on this topic, signifies the significance of this issue in contemporary research. This observation underscores the recognition of the potentially adverse effects that real estate bubbles can have on economies and societies.

Despite the dynamic nature of knowledge and the rapid advancements in various fields, it is intriguing to note the limited thematic variations observed within the research on real estate bubbles. The lack of major innovations in thematic approaches, as illustrated by the dominance of consumption as the only emerging topic in Figure 5, raises questions about the extent of thematic conformity within the field. The absence of artificial intelli-

gence, machine learning, and smart cities as new approaches suggest that these innovative methodologies may not have gained significant traction or recognition among experts studying real estate bubbles.

Furthermore, the study reveals the presence of three distinct bodies of production based on their approaches. The financial approach seeks to understand the rationality behind real estate bubble formation and influence decision-makers to prevent market collapses. The methodological approach focuses on enhancing the accuracy of measuring real estate bubbles through empirical techniques. Lastly, the social approach delves into the social causes and consequences of real estate bubbles, with a particular emphasis on social justice and equity. While these approaches individually contribute valuable insights, the limited interaction among them highlights the potential for a more integrated and interdisciplinary research framework.

Additionally, the analysis demonstrates a tendency for certain nations to collaborate with each other in producing research papers on housing bubbles. The categorization of nations into groups emphasizing liberal approaches, welfare states, and local or less associated approaches, as depicted in Figure 7, suggests potential associations between research orientation and broader socio-political perspectives. Further investigation and separate bibliometric analyses of these groups may offer deeper insights into the variations in research productivity and perspectives on real estate bubbles.

The study's findings also emphasize the need for greater attention to be given to Latin America and Africa, regions where housing crises and high prices persist. The absence of dominant studies from these regions presents an opportunity for researchers and policymakers to explore real estate bubbles as a dedicated field of study and formulate effective strategies to address housing challenges.

Lastly, the study underscores the importance of promoting thematic and methodological innovation in research on real estate bubbles. The limited innovation observed in the most influential studies indicates the potential for advancements in incorporating artificial intelligence, machine learning, and other emerging technologies to enhance monitoring and prevention mechanisms. The utilization of automated systems and the existing technological capabilities can significantly contribute to mitigating the risks associated with real estate bubbles.

In conclusion, the bibliometric analysis conducted in this study sheds light on the scientific productivity, research trends, and areas for improvement in the study of real estate bubbles. The insights gained from this analysis can guide future research endeavors, promote interdisciplinary collaboration, and facilitate the development of innovative approaches to understanding and addressing the challenges posed by real estate bubbles. By continually advancing the knowledge and methodologies in this field, policymakers, researchers, and stakeholders can work together to foster sustainable and resilient real estate markets that contribute positively to economic growth and social well-being.

**Funding:** This research received no external funding and The APC was funded by Vicerrectoría de Investigación of Universidad de Las Américas, Chile, GRANT 142.

**Informed Consent Statement:** Not applicable.

**Data Availability Statement:** The data presented in this study are available on request from the corresponding author. The data are not publicly available due to privacy and in order to track who is using the data to run new analyses. Just ask for it.

**Conflicts of Interest:** The author declares no conflict of interest.

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
