# Peer review of "A Systematic Bibliometric Analysis of the Real Estate Bubble Phenomenon: A Comprehensive Review of the Literature from 2007 to 2022"

_ijfs, doi:10.3390/ijfs11030106_

Round 1

Reviewer 1 Report

The paper is of interest, especially to real estate researchers.  Some figures are too small to read, and I do not see the purpose of Figure 2.  Pages 5-6 (trend topics) were far too small to see. I am not a fan of word clouds, but I do see them in qualitative literature.  The author appears to be a singular person yet writes as ''we." In the end, the paper was well written but left me with a 'what now?' question.  That is, it did not read as strongly motivated except for following a series of processes.  I am sure this is not the intent of the author, but it is how it came across to me.  I recommend revising for resubmission with encouragement.

Overall this is good except question use of 'we'

Author Response

Response to Reviewer 1

We would like to express our sincere gratitude for your valuable time and insightful observations on our manuscript titled "A Bibliometric Analysis of the Real Estate Bubble in Literature between 2007 and 2022." We highly appreciate your efforts in critically evaluating our work and providing constructive comments to enhance the quality and impact of our research.

To address your observations, we have carefully reviewed each comment and revised our manuscript accordingly. In this response, we will address your comments point-by-point, outlining the revisions made and the underlying rationale behind our decisions.

COMMENTS & RESPONSES

The paper is of interest, especially to real estate researchers. 

RESPONSE: Thanks for this comment.

Some figures are too small to read, and I do not see the purpose of Figure 2. Pages 5-6 (trend topics) were far too small to see.

RESPONSE: Thanks for this observation. Figures have been revised and rendered in order to increase its readability.

In the end, the paper was well written but left me with a 'what now?' question.  That is, it did not read as strongly motivated except for following a series of processes.  I am sure this is not the intent of the author, but it is how it came across to me.  I recommend revising for resubmission with encouragement.

RESPONSE: Thanks for this observation. Several sections of the manuscript has been revised in order to deliver a deeper analysis and make it clearer the relevance of the findings for studying this topic.

Reviewer 2 Report

The aim of the paper is to present the results of a bibliometric review of studies on real estate bubbles in the scientific literature indexed in Web of Science and Scopus from 2007 to 2022.

The paper addresses an interesting and particularly topical subject with adequate clarity.

However, it is considered necessary to introduce some revisions.

In order to keep the introduction comprehensible to scientists working outside the topic of the paper it is suggested 

- to specify what is meant and which 'Fundamentals' it is about (line 20) ;

- to make the description of the structure of the article congruent with its actual organisation of the parts [The fourth Section of the paper, in its current drafting, does not present the results of the analysis and conclusions, as is indicated in lines 42-43, but rather materials and methods, line 264; the second section presents the results of the investigations carried out and the third Section the discussion of the results obtained]. The description of the structure of the paper could also be moved to the end of Section 1

In Section 3 it could be useful (also to corroborate the statements in lines 258-262: "Low thematic or methodological innovation has been identified in most of the studies, or at least in the most influential ones. The absence of artificial intelligence or machine learning techniques or both together may well be at an early stage, but it seems entirely appropriate to try to generate automated mechanisms to monitor and prevent the emergence of bubbles") specify the methodologies and tools adopted and used in each of the reviewed papers relating to the indicated approaches (lines 232-239); this can also be done by developing a specific elaboration of the bibliometric data under investigation to be described and included in Section 2.

 Finally, it is suggested to extend the conclusions 

Author Response

Response to Reviewer 2

We would like to express our sincere gratitude for your valuable time and insightful observations on our manuscript titled "A Bibliometric Analysis of the Real Estate Bubble in Literature between 2007 and 2022." We highly appreciate your efforts in critically evaluating our work and providing constructive comments to enhance the quality and impact of our research.

To address your observations, we have carefully reviewed each comment and revised our manuscript accordingly. In this response, we will address your comments point-by-point, outlining the revisions made and the underlying rationale behind our decisions.

COMMENTS & RESPONSES

- to specify what is meant and which 'Fundamentals' it is about (line 20) ;

RESPONSE: Thanks for this comment. It was clarified in the same sentence.

- to make the description of the structure of the article congruent with its actual organisation of the parts [The fourth Section of the paper, in its current drafting, does not present the results of the analysis and conclusions, as is indicated in lines 42-43, but rather materials and methods, line 264; the second section presents the results of the investigations carried out and the third Section the discussion of the results obtained]. The description of the structure of the paper could also be moved to the end of Section 1

RESPONSE: Thanks for this observation. This part of the introduction was amended according to these observations.

In Section 3 it could be useful (also to corroborate the statements in lines 258-262: "Low thematic or methodological innovation has been identified in most of the studies, or at least in the most influential ones. The absence of artificial intelligence or machine learning techniques or both together may well be at an early stage, but it seems entirely appropriate to try to generate automated mechanisms to monitor and prevent the emergence of bubbles") specify the methodologies and tools adopted and used in each of the reviewed papers relating to the indicated approaches (lines 232-239); this can also be done by developing a specific elaboration of the bibliometric data under investigation to be described and included in Section 2.

RESPONSE: Thanks for this observation. Discussion and Conclusions were reversed in order to add more specificities of methods used.

Finally, it is suggested to extend the conclusions

RESPONSE: The whole section of conclusions was extended and revised where are included new responses to these observations too.

Reviewer 3 Report

See attached

I have put my suggestions for proofreading grammatical errors in the attached referee report. 

Author Response

Response to Reviewer 3

We would like to express our sincere gratitude for your valuable time and insightful observations on our manuscript titled "A Bibliometric Analysis of the Real Estate Bubble in Literature between 2007 and 2022." We highly appreciate your efforts in critically evaluating our work and providing constructive comments to enhance the quality and impact of our research.

To address your observations, we have carefully reviewed each comment and revised our manuscript accordingly. In this response, we will address your comments point-by-point, outlining the revisions made and the underlying rationale behind our decisions.

COMMENTS & RESPONSES

RESPONSE TO FIRST COMMENT: Thanks for this comment, it encouraged on further analysis for new research as such. It was clarified in new paragraphs added to the document.  

RESPONSE TO SECOND COMMENT: This was a comment similar to another reviewer's. Indeed, in the pursuit of generating a neutral wording of the text, the richness of the argumentation was lost. In this new version, efforts have been made to improve the presentation of the arguments and appreciate the results. Regarding the search for the specificity of each database by separating them based on methodologies, it is a task that is not part of the objectives of this research and requires a different approach to the documents, as indicated in the methods section. The focus here is on a comprehensive analysis rather than the particularities of each article, which would be more of a text mining study rather than bibliometrics.

RESPONSE TO THIRD COMMENT: Thank you very much for these contributions. They have been taken into consideration to make some changes, as they are consistent with the observations made by other reviewers as well.

RESPONSE TO FOURTH COMMENT: Indeed, this issue was present in the previous version, with insufficient development of the conclusions. The new version adds substantial content in this regard, both in the discussion and the actual conclusions.

RESPONSE TO LAST COMMENT: The text has undergone thorough proofreading, and the observed errors have been corrected. Thank you again

Reviewer 4 Report

Thank you for allowing me to read your paper. PLEASE NOTE THE FOLLOWING COMMENTS before I can recommend your paper for publication. Please respond to each comment in detail and indicate the changes in your new revised manuscript. Give specific reference to the page and line number in your response.

1.     Please rewrite your title, especially the word "between." Here is a suggestion: "A Systematic Bibliometric Analysis of the Real Estate Bubble Phenomenon: A Comprehensive Review of the Literature from 2007 to 2022" OR "Bibliometric Analysis of the Real Estate Bubble: A Comprehensive Review of Literature from 2007 to 2022"

2.     Despite the widespread recognition and reliability of Scopus and WOS as databases, variations exist in the calibre of information furnished by each. It is plausible that data entry errors, duplications, or inaccuracies could potentially impact the outcomes of the analysis. Thoroughly examining and purifying data is a crucial measure in mitigating bias. Data pre-processing is a necessary step that needs to be undertaken in this study. The methodology section lacks the information I am seeking. Effective pre-processing techniques, such as deduplication, can guarantee the utilisation of precise and untainted data for bibliometric analysis, leading to the generation of research outcomes that are more significant and pertinent. To clarify my concern I like to emphasize that  while Scopus and Web of Science (WOS) are widely recognized and reputable databases, it is crucial to acknowledge that variations exist in the quality of data provided by each platform. Consequently, the potential for errors, duplications, or inaccuracies in data entry emerges, subsequently impacting the integrity of the analysis outcomes. Therefore, undertaking meticulous data pre-processing is essential to mitigate any potential biases. It is important to note that the methodology presented in the study lacks explicit details on data pre-processing procedures. By implementing robust pre-processing techniques, including comprehensive deduplication measures, the research can ensure that the bibliometric analysis is conducted using accurate and refined data. This, in turn, enhances the validity and relevance of the research findings

3.     The utilization of two prominent databases, namely Scopus and Web of Science (WOS), adds value to this study. However, in the "Discussion" section, it is essential to provide clarity regarding the specific database being discussed. Specifically, it is recommended to explicitly state the number of articles retrieved from each database, allowing readers to comprehend the scale and breadth of the research. Moreover, elucidating the approach employed to analyze and interpret the data from both databases would significantly enhance the transparency and credibility of the study. By elaborating on the methodology used to handle the data from Scopus and WOS, including any potential variations or challenges encountered, the readers can gain deeper insights into the analytical process and better evaluate the robustness of the findings. Therefore, it is imperative to address these aspects in the "Discussion" section to provide a comprehensive overview of the database selection, article count, and the subsequent analytical procedures undertaken for both Scopus and WOS datasets.

4.     Typo Line 23. The search results yielded 2496 results ranging from 1979 to 20223?

5.     The research paper's title, which reads "A bibliometric analysis of the real estate bubble in literature between 2007 and 20223," (Line 247) seems to be at odds with the findings of the search, which cover the period from 1979 all the way up to 2023 or 2022.

6.     The R package offers immense flexibility for conducting statistical analysis and tailor-made data processing, rendering it a valuable tool in various research domains. However, in the realm of bibliometric analysis, VOSviewer emerges as a more widely adopted software with notable advantages in terms of interactive visualization, citation analysis, and seamless data integration. As a result, researchers often employ a combination of both R and VOSviewer to augment their analyses and attain a more comprehensive understanding of bibliometric data.

Considering the popularity and advantages of VOSviewer, it is essential to address the reasons for exclusively utilizing R programs in this study. By elucidating the rationale behind this decision, the study can provide transparency and acknowledge potential limitations. It is crucial to acknowledge that utilizing only R programs may limit the researchers' ability to leverage VOSviewer's interactive visualization capabilities and advanced citation analysis features, which could potentially lead to a narrower perspective on the bibliometric landscape.

Therefore, to enhance the comprehensiveness and potential insights derived from the study, it is recommended to clarify the specific reasons and justifications for exclusively using R programs. By doing so, readers can better understand the study's limitations and assess the potential impact on the overall findings and interpretations.

Author Response

Response to Reviewer 4

We would like to express our sincere gratitude for your valuable time and insightful observations on our manuscript titled "A Bibliometric Analysis of the Real Estate Bubble in Literature between 2007 and 2022." We highly appreciate your efforts in critically evaluating our work and providing constructive comments to enhance the quality and impact of our research.

To address your observations, we have carefully reviewed each comment and revised our manuscript accordingly. In this response, we will address your comments point-by-point, outlining the revisions made and the underlying rationale behind our decisions.

COMMENTS & RESPONSES

  1. Please rewrite your title, especially the word "between." Here is a suggestion: "A Systematic Bibliometric Analysis of the Real Estate Bubble Phenomenon: A Comprehensive Review of the Literature from 2007 to 2022" OR "Bibliometric Analysis of the Real Estate Bubble: A Comprehensive Review of Literature from 2007 to 2022"

RESPONSE 1: Thanks for this generous suggestion. The new title is: A Systematic Bibliometric Analysis of the Real Estate Bubble Phenomenon: A Comprehensive Review of the Literature from 2007 to 2022

  1. Despite the widespread recognition and reliability of Scopus and WOS as databases, variations exist in the calibre of information furnished by each. It is plausible that data entry errors, duplications, or inaccuracies could potentially impact the outcomes of the analysis. Thoroughly examining and purifying data is a crucial measure in mitigating bias. Data pre-processing is a necessary step that needs to be undertaken in this study. The methodology section lacks the information I am seeking. Effective pre-processing techniques, such as deduplication, can guarantee the utilisation of precise and untainted data for bibliometric analysis, leading to the generation of research outcomes that are more significant and pertinent. To clarify my concern I like to emphasize that while Scopus and Web of Science (WOS) are widely recognized and reputable databases, it is crucial to acknowledge that variations exist in the quality of data provided by each platform. Consequently, the potential for errors, duplications, or inaccuracies in data entry emerges, subsequently impacting the integrity of the analysis outcomes. Therefore, undertaking meticulous data pre-processing is essential to mitigate any potential biases. It is important to note that the methodology presented in the study lacks explicit details on data pre-processing procedures. By implementing robust pre-processing techniques, including comprehensive deduplication measures, the research can ensure that the bibliometric analysis is conducted using accurate and refined data. This, in turn, enhances the validity and relevance of the research findings

RESPONSE 2: Thanks for this request, which actually was run before the data analysis. A new paragraph was added to the section five in order to clarify this point, which is read as follows:

Although Scopus and Web of Science (WoS) are widely recognized and reputable databases, it is crucial to acknowledge that there are variations in the quality of data provided by each platform. It should be noted that some journals are indexed in both databases, which can lead to potential errors, duplications, or inaccuracies in the data. These issues can ultimately impact the integrity of the analysis results. Therefore, to address this concern, a data pre-processing step was implemented, which involved comprehensive deduplication using R and the DPLYR package. Subsequently, a visual review of the data cleansing was conducted. As part of this pre-processing stage, approximately 830 duplicated records were eliminated from the sample.

  1. The utilization of two prominent databases, namely Scopus and Web of Science (WOS), adds value to this study. However, in the "Discussion" section, it is essential to provide clarity regarding the specific database being discussed. Specifically, it is recommended to explicitly state the number of articles retrieved from each database, allowing readers to comprehend the scale and breadth of the research. Moreover, elucidating the approach employed to analyze and interpret the data from both databases would significantly enhance the transparency and credibility of the study. By elaborating on the methodology used to handle the data from Scopus and WOS, including any potential variations or challenges encountered, the readers can gain deeper insights into the analytical process and better evaluate the robustness of the findings. Therefore, it is imperative to address these aspects in the "Discussion" section to provide a comprehensive overview of the database selection, article count, and the subsequent analytical procedures undertaken for both Scopus and WOS datasets.

RESPONSE 3: Thanks for this observation. This clarification was included in the method section, but also added new sentences on this matter on conclusions and discussion sections.

  1. Typo Line 23. The search results yielded 2496 results ranging from 1979 to 20223?

RESPONSE 4: Thanks, it was amended.

  1. The research paper's title, which reads "A bibliometric analysis of the real estate bubble in literature between 2007 and 20223," (Line 247) seems to be at odds with the findings of the search, which cover the period from 1979 all the way up to 2023 or 2022.

RESPONSE 5: Thanks for this observation. The article collected papers from 1979 to 2023. However, the aim was to trace the effects on literature from the bubble of 2008 to 2022, as 2023 is an incomplete year. This clarification was added to the final section of the manuscript.

  1. The R package offers immense flexibility for conducting statistical analysis and tailor-made data processing, rendering it a valuable tool in various research domains. However, in the realm of bibliometric analysis, VOSviewer emerges as a more widely adopted software with notable advantages in terms of interactive visualization, citation analysis, and seamless data integration. As a result, researchers often employ a combination of both R and VOSviewer to augment their analyses and attain a more comprehensive understanding of bibliometric data.

RESPONSE 6: Thanks for this suggestion, however, this articles method uses R only and it would imply a methodological change to shift to VOSviewer, which cannot perform other type of analysis such as those presented in this article.

Reviewer 5 Report

I have a few recommendations:

1. the paper is a review, not an article. It should be changed

2. Between Introduction and Results there should be a section named Methodology in which you should explain the steps, the software used and the entire process of collecting data.

3. Quality of the figures should be improved. In Figure 1, the writing on the X-axis is not clear. 

4. Line 11, after Table 1 there should be space before the verb indicates

5. Tables should have the same format (font dimension)

6. You have Materials and Methods after Results and Discussion. You should move the section after the introduction. Reorder figures and tables if it affects them

7. You need to add a Conclusion section at the end to include the following: Theoretical and Practical implications of your research, what is your contribution to the field? the novelty of your paper. Also in Conclusions, you should add a paragraph about the Limitations of your study (related to the method and the papers excluded from the research) and Future Research Directions (related to some gaps in the literature review). 

8. For a review paper and bibliometric analysis, the number of references is too low. Please add more references to your paper. You can add them in the introduction or Discussions and Conclusions. 

Author Response

Response to Reviewer 5

We would like to express our sincere gratitude for your valuable time and insightful observations on our manuscript titled "A Bibliometric Analysis of the Real Estate Bubble in Literature between 2007 and 2022." We highly appreciate your efforts in critically evaluating our work and providing constructive comments to enhance the quality and impact of our research.

To address your observations, we have carefully reviewed each comment and revised our manuscript accordingly. In this response, we will address your comments point-by-point, outlining the revisions made and the underlying rationale behind our decisions.

COMMENTS & RESPONSES

  1. the paper is a review, not an article. It should be changed

RESPONSE 1: Thanks. I recall to the editors to make this change that I agree as I cannot make it from the submission platform.

  1. Between Introduction and Results there should be a section named Methodology in which you should explain the steps, the software used and the entire process of collecting data.

RESPONSE 2: I have followed the journal’s template which demands this type of structure. I am not able to change that requirement which emerged from the guidelines of the IJFS.

  1. Quality of the figures should be improved. In Figure 1, the writing on the X-axis is not clear. 

RESPONSE 3: Thanks for this observation. Figures were amended.

  1. Line 11, after Table 1 there should be space before the verb indicates

RESPONSE 4: Thanks for this observation. A throughout revision and proofreading of the manuscript was run.

  1. Tables should have the same format (font dimension)

RESPONSE 5: Thanks for this observation, tables were accommodated within the space available and fonts tried to become unified. I recall the journal style designers to amend this problem in order to find a solution suitable to their interal aesthetic policies.

  1. You have Materials and Methods after Results and Discussion. You should move the section after the introduction. Reorder figures and tables if it affects them

RESPONSE 6: As I said in response 2, it is part of the journal’s guidelines.

  1. You need to add a Conclusion section at the end to include the following: Theoretical and Practical implications of your research, what is your contribution to the field? the novelty of your paper. Also in Conclusions, you should add a paragraph about the Limitations of your study (related to the method and the papers excluded from the research) and Future Research Directions (related to some gaps in the literature review). 

RESPONSE 7: Thanks, a whole new conclusion section was added separately from the discussion section.

Round 2

Reviewer 3 Report

See attached. 

NA. 

Author Response

COMMENTS & RESPONSES REVIEWER 3

In the revised manuscript, I did not find sufficient responses to my previous comments, nor was there any explanation provided for the absence of responses.

RESPONSE: All comments from the first round have been addressed. In the peer review process, there is no obligation to accept all comments from reviewers. However, it is crucial to respond to those comments in order to establish a peer conversation about the foundations, contributions, methods used, and the decisions made by the authors prior to the provided observations. It is possible that these responses may contradict the original comments, but as previously mentioned, this is a normal part of the peer review process for reviewing articles. It is important to note that four other reviewers of this paper have different views on the observations. Reconciling the differences among the provided observations is also a challenging task to address. Nevertheless, please, have a closer look to the new version of the paper as new amendments were included.

The authors did not follow my suggestions to enhance or clarify the tables and figures, except for the capitalization of three words in Figure 4.

RESPONSE: Please take another look as all the figures have been recreated and enhanced in Photoshop to improve readability. The initial version of this paper included images directly from the R software, so the enhanced figures are actually much better.

Moreover, the revised paper fails to demonstrate significant improvements in terms of explaining the existence of the literature gap in Latin America, providing motivation for suggesting utilizing machine learning or artificial intelligence, and summarizing the prevailing explanation for the housing crisis from the reviewed studies.

RESPONSE: Thank you for this observation. Several sections of the manuscript have been revised to provide a more in-depth analysis and to clarify the relevance of the findings in the study of this topic.

Furthermore, as a literature review, this paper lacks comprehensive coverage of relevant studies.

RESPONSE: Thank you for pointing this out. New references have been added. However, it is important to mention that some of the references provided by the reviewer would be useful for a literature review study on real estate bubbles that applies a thematic analysis of sources. This research focuses on a bibliometric analysis, and if the provided references were not included in the analysis, it is mainly because they have not been cited enough and therefore have not achieved relative impact compared to the total sample of articles that do appear.

Reviewer 4 Report

All responses are satisfactory except response 6.

"RESPONSE 6: Thanks for this suggestion, however, this articles method uses R only and it would imply a methodological change to shift to VOSviewer, which cannot perform other type of analysis such as those presented in this article."

I suggest the author(s) incorporate a paragraph or at least a few lines making their points with documentation and proper citation.  

Also, author(s) might benefit by looking at this paper:

Moslehpour, M.; Shalehah, A.; Rahman, F.F.; Lin, K.-H. The Effect of Physician Communication on Inpatient Satisfaction. Healthcare 202210, 463. https://doi.org/10.3390/healthcare10030463

Author Response

All responses are satisfactory except response 6. "RESPONSE 6: Thanks for this suggestion, however, this articles method uses R only and it would imply a methodological change to shift to VOSviewer, which cannot perform other type of analysis such as those presented in this article."

I suggest the author(s) incorporate a paragraph or at least a few lines making their points with documentation and proper citation. 

RESPONSE: Thank you for highlighting this observation. I agree that more details are needed. VosViewer is widely used software for this type of analysis, and the use of bibliometrix (which already includes most of the mapping tools similar to VOSviewer) requires further explanation to properly convey the advantages of using only bibliometrix for conducting the analyses. A new paragraph has been added to expand on this concept.

There are some advantages of using the bibliometrix package. It provides a flexible framework for bibliometric analysis, allowing researchers to customize and adapt the way of conducting the study according to their specific research questions and needs. It offers a wide range of functions and parameters that can be tailored to the analysis objectives such as data collection and preprocessing in the R environment to tidy up the datasets. Also, the package offers network analysis capabilities, allowing researchers to analyze collaboration networks, co-citation networks, and co-word networks, not only as visualization as in VOSviewer, but also based on its indicators and functions to calculate network centrality measures, identify clusters or communities within the network, and visualize network structures in a similar way than in VOSviewer. Furthermore, the use of temporal analysis enables researchers to examine temporal trends in bibliometric data. It offers functions to identify prolific authors, assess author productivity and collaboration patterns, and explore the research performance of institutions, and also use keywords and abstract n-grams to elaborate more complex analysis near to text-mining tools (which may also be incorporated from the R environment although it was not applied to this research particularly). So, as a package in R, bibliometrix seamlessly integrates with other R packages and tools, enabling researchers to incorporate bibliometric analysis into their existing R workflows. This integration facilitates data preprocessing, statistical analysis, and visualization, as R offers a comprehensive ecosystem of packages for these tasks. Finally, bibliometrix facilitates the automatation of the entire bibliometric analysis process, from data collection to visualization, using R scripts. This enables reproducibility, as the analysis can be easily replicated and shared with others, enhancing transparency and facilitating collaboration.

Also, author(s) might benefit by looking at this paper:

Moslehpour, M.; Shalehah, A.; Rahman, F.F.; Lin, K.-H. The Effect of Physician Communication on Inpatient Satisfaction. Healthcare 2022, 10, 463. https://doi.org/10.3390/healthcare10030463

RESPONSE: Thank you for this recommendation. This suggestion is related to a previous suggestion made in the first round. Indeed, it helped to better understand the meaning of some of the previous reviews.

Round 3

Reviewer 3 Report

I do not have any other comments.